# Theoretical description and experimental simulation of quantum entanglement near open time-like curves via pseudo-density operators

Chiara Marletto [1,2,3], Vlatko Vedral[1,2,3,4], Salvatore Virzì[5,6], Enrico Rebufello[6,7], Alessio Avella[6], Fabrizio Piacentini[6], Marco Gramegna[6], Ivo Pietro Degiovanni [6] & Marco Genovese [8]

Closed timelike curves are striking predictions of general relativity allowing for time-travel. They are afflicted by notorious causality issues (e.g. grandfather's paradox). Quantum models where a qubit travels back in time solve these problems, at the cost of violating quantum theory's linearity—leading e.g. to universal quantum cloning. Interestingly, linearity is violated even by open timelike curves (OTCs), where the qubit does not interact with its past copy, but is initially entangled with another qubit. Non-linear dynamics is needed to avoid violating entanglement monogamy. Here we propose an alternative approach to OTCs, allowing for monogamy violations. Specifically, we describe the qubit in the OTC via a pseudo-density operator—a unified descriptor of both temporal and spatial correlations. We also simulate the monogamy violation with polarization-entangled photons, providing a pseudo-density operator quantum tomography. Remarkably, our proposal applies to any space-time correlations violating entanglement monogamy, such as those arising in black holes.

[1] Clarendon Laboratory, University of Oxford, Parks Road, Oxford OX1 3PU, UK. [2] Fondazione ISI, Via Chisola 5, Torino 10126, Italy. [3] Centre for Quantum Technologies, National University of Singapore, 3 Science Drive 2, Singapore 117543, Singapore. [4] Department of Physics, National University of Singapore, 2 Science Drive 3, Singapore 117542, Singapore. [5] Università di Torino, via P. Giuria 1, 10125 Torino, Italy. [6] Istituto Nazionale di Ricerca Metrologica, Strada delle Cacce 91, 10135 Torino, Italy. [7] Politecnico di Torino, Corso Duca degli Abruzzi 24, 10129 Torino, Italy. [8] INFN – sezione di Torino, Via P. Giuria, 110125 Torino, Italy. Correspondence and requests for materials should be addressed to C.M. (email: chiara.marletto@gmail.com)

Quantum theory and general relativity each provide well-verified predictions, in their respective domains. However, they also provide predictions that cannot yet be probed experimentally, but give one the opportunity of exploring physics which is rather different from what we perceive directly at our scales. Of particular interest are predictions of space-time correlations violating the standard properties of quantum theory, such as superpositions of different space-time geometries in quantum gravity, resulting in superposing different causal orders[1], or the physics of black holes[2]. In these cases, it is possible to relax some of the assumptions of quantum theory and still have a coherent picture—which leads to proposals for new frameworks that go beyond quantum theory. An important example of such violations is the dynamics of a quantum system near closed timelike curves (CTCs). CTCs are allowed solutions of Einstein's equations, which provide a model for time travel: they allow observers to travel backwards in time and, possibly, even to interact with their former selves. These solutions have been argued to be unphysical in classical general relativity, because they lead to paradoxes, such as the grandfather's paradox[3,4]. Some even invoke a chronology-protection principle to rule out their existence in physical reality[5]. Another possible resolution of the paradoxes, however, comes unexpectedly from merging general relativity with quantum theory, by considering the dynamics of a quantum object (e.g. a qubit) going back in time through a CTC and interacting with its past copy[6] (see also refs. [7–11] for recent developments.). Although the classical paradoxes seem to be resolved within this approach, the resulting dynamical evolution on each of the qubit copies is non-linear[6]. Because of non-linearity, CTCs can be used to perform perfect discrimination of non-orthogonal states and other tasks that violate quantum theory[12–15]. This non-linear evolution has also been experimentally simulated[9]. Interestingly, even when there is no interaction between the earlier and later copies of the qubit, that is, when there is an open timelike curve (OTC), there can be violations of basic properties of entanglement, if the qubit is initially entangled with another, chronology-respecting, qubit[14,16]. Although monogamy of entanglement is violated in the chronology-violating region, verifying the violation seems practically hard, because it would require to act on the qubit entering the open timelike curve, which would affect the state of the qubit itself. The usual approach to an OTC, which preserves monogamy, is to assume that the resulting dynamics on the subsystems in the OTC regions violates unitarity by being entanglement breaking.

Here instead we propose an alternative solution: that the state describing the chronology-violating region is not a density operator. This, as we shall explain, allows one to describe the overall state of the chronology-violating region, maintaining that the monogamy of entanglement is violated. Here we shall focus on the original model proposed by Deutsch[6], where a qubit interacts unitarily with a copy of itself that is sent back in time, via the CTC. Extending our proposed framework to alternative models such as those resorting to post selection[10] is an interesting development which we leave for a future paper. Our proposal consists of two parts. First, in order to describe the state of the qubits in the chronology-violating region, we resort to the recently proposed tool called pseudo-density operator (PDO)[17]. PDOs were originally introduced to treat quantum correlations in space and time on an equal footing; they are Hermitian, trace-one operators, which are not necessarily positive, and therefore can describe time as well as space correlations[17]. Our proposal is a new application of PDOs, to describe the state of a qubit that violates monogamy of entanglement because it enters an OTC. As we shall see, our approach allows one to preserve linearity in an interesting way—because any two different PDOs are related by a

linear transformation. This opens a new line of investigation where instead of modifying the linearity of quantum theory we modify other properties, specifically the positivity of the quantum state, to accommodate features induced by other physical requirements, in this case general relativity. An interesting application of this work would be to consider how other approaches to incorporate space-time correlations in quantum theory[18] could be used to the same effect as the PDO in this context, and also to explore how the CTC scenario would be describable in this approach. Although presented for the OTC scenario, this approach is very general and could be adapted to other cases that seemingly violate quantum theory in the same way, such as the black hole entropy paradox[19,20]. The second part of our proposal is an experimental demonstration of the statistics of the OTC, where we simulate the entanglement monogamy violation and provide a full tomographic reconstruction of the whole PDO. This sets the paradigm for the experimental reconstruction of the PDO, which as we shall explain presents interesting subtleties. The simulation of the OTC consists in reproducing the correlations in the PDO that we conjecture can describe the OTC.

## Results

**A model with PDOs.** In quantum theory, the complete specification of the state of a physical system is given at any one time by its density operator, and the initial conditions; the density operator of a composite system contains all the possible correlations between its subsystems. A PDO generalises density operators to include temporal correlations between systems measured at multiple times, thereby treating the tensor product as both combining spacelike or timelike separated systems. We note that a similar formal tool was already introduced by Isham[21] in the context of the consistent-history approach. For a review of the formal properties of the PDO see ref. [18]. Here we shall use an example, to understand what a PDO means physically. Consider the statistics of a physical process where a single qubit, initially in a maximally mixed state, is measured at two different times. Each measurement could be performed in any of the three complementary bases $X$, $Y$, $Z$ (represented by the usual Pauli operators—the choice of basis is, as always, arbitrary). Suppose we would like to write those statistics in the form of an operator, generalising the quantum density operator. Because the whole state, as we shall see, is Hermitian and unit trace, but not positive, we call it 'pseudo-density matrix'.

It is represented as:

$$R_{12} = \frac{1}{4}\{I_{12} + X_1 X_2 + Y_1 Y_2 + Z_1 Z_2\}, \tag{1}$$

where 1 and 2 represent two different times. This operator has similarities with the density operator of a singlet state; however, the correlations all have a positive sign, whereas for the singlet they are all negative, $\langle XX \rangle = \langle YY \rangle = \langle ZZ \rangle = -1$. In fact, it is simple to show that $R_{12}$ is not a density operator, because it is not positive (i.e. it has at least one negative eigenvalue). We can, however, trace the label 2 out and obtain one marginal, that is, the 'reduced' state of 1. Interestingly, this itself is a valid density matrix, corresponding to the maximally mixed state $I/2$. Likewise for the subsystem 2. So, the marginals of this generalised operator are actually both perfectly allowed physical states, but the overall state is not.

As a result of the presence of temporal correlations, a PDO is not necessarily a positive operator, although it still is trace-one and Hermitian. This means that it presents negative expectation values of projectors. For example, $R_{12}$ has the singlet state as an

eigenstate, with eigenvalue $-\frac{122}{2}$. This is interpreted as the signature of correlations in time[23].

In our paper we propose a different application of PDOs: as a generalised state which can describe the statistics of the system consisting of a qubit entering an OTC, its future copy emerging from it, and another qubit that is maximally entangled with it. This state encapsulates the violation of the monogamy of quantum entanglement that is caused in the OTC region, and provides a full consistent description for the three-qubit system within the chronology-violating region. This is different from other proposals, where the monogamy of entanglement is preserved at the expenses of introducing a non-linear evolution. Here, we conjecture that the state $R_{12}$ presented above describes the joint state of the qubit that is sent back in time via an OTC, and its copy that emerges from the OTC. This is because the two qubits are then perfectly correlated in all bases. Interestingly, as we said, this description can be thought of as preserving linearity because any two PDOs of the same dimensionality are Hermitian operators and thus can be related to one another via a linear transformation. Let us now introduce our model in more detail. A maximally entangled pair of qubits (Q1 and Q2) is created in the distant past of the region of spacetime that contains the OTC; qubit Q2 is then sent into the OTC. Let the copy emerging from the OTC be represented by a third qubit (qubit Q3). In the distant past and the distant future, the state of the qubits is just a maximally entangled pair. However, in the chronology-violating region we describe the whole state as a PDO $R_{123}$, which represents the fact that Q1 has to be maximally entangled both with the qubit that emerges from the OTC (Q3) and with the qubit entering it (Q2)—see Fig. 1. The two marginal PDOs $R_{12}$ and $R_{13}$ are two density operators, representing each a maximally entangled pair; the marginal $R_{23}$ is, instead, a PDO (not a physical state) describing perfect correlation in time between the past copy of the qubit and the future copy; the whole descriptor $R_{123}$ is also not a physical state.

The total PDO describing the chronology-violating region can be written as

$$R_{123} = \frac{1}{8}\{I_{123} - \Sigma_{12} + \Sigma_{23} - \Sigma_{13})\}, \qquad (2)$$

where $\Sigma_{ij} = X_iX_jI_k + Y_iY_jI_k + Z_iZ_jI_k$ and $I_k$ is the unit on system $k$. The reduced states are $R_{23} = \frac{1}{4}(I_{23} + \Sigma_{23})$, $R_{12} = \frac{1}{4}(I_{12} - \Sigma_{12})$ and $R_{13} = \frac{I}{4}(I_{13} - \Sigma_{13})$. Now we can see that qubits Q1 and Q2 can be maximally entangled, as they were prepared as such in the distant past; Q1 and Q3 can also be maximally entangled, because qubit Q3 is the copy of qubit Q2 that entered the OTC; while Q2

and Q3 are maximally correlated in all bases, because they describe the later and earlier qubits in the chronology-violating region, they are therefore described by a PDO, which is not a physical state. The overall PDO describes a state where again three qubits are maximally anti-correlated in every basis, which is an unphysical state. Note also the subtlety that the qubit entering an OTC could undergo some unitary transformation. This transformation would not change its being maximally entangled with the other qubit, so it could be incorporated in the description above by modifying the reduced state of Q1 and Q2 and of Q2 and Q3 to be different maximally entangled states. However, it still remains true that the qubit just before entering the OTC (Q2) and just after emerging from it (Q3) are two copies of the same qubit, which is why they can be described by the PDO $R_{23}$.

**Monogamy violation.** Experimentally speaking, one of the simplest ways of testing the violation of entanglement monogamy is to use the violation of Bell's inequalities (whose violation is sufficient to witness the existence of entanglement). Specifically, setting $C_{ij} = Tr(R_{ij} \ B_{ij})$, where $B_{ij} = \sqrt{2}(X_iZ_j + Z_iX_j)$ is the observable that is used in the Clauser–Horne–Shimony–Holt (CHSH) inequality tests on qubits $i$, $j$, one has[24]:

$$C_{mk} + C_{nk} \leq 4. \qquad (3)$$

In other words, for quantum states of three qubits $m$, $n$, $k$, we cannot violate Bell's inequalities in more than one pair of qubits.

One can show that this inequality is violated in the state described by $R_{123}$. Since $R_{12}$ and $R_{13}$ describe each a maximally entangled pair, $C_{12} = 2\sqrt{2} = C_{13}$, the former inequality is violated. The same for $R_{12}$ and $R_{23}$, given that the latter also describes perfect correlations in all basis, $C_{12} = 2\sqrt{2} = C_{23}$. Note that this is a different application of the PDO to describe two distinct timelike separated qubits, that is, the past and future copy of the qubit within the OTC, which are perfectly correlated in all bases. This is different from the standard use of the PDO as a tool to describe timelike correlations (which are already known to violate monogamy of entanglement when considering the time evolution of a single qubit[25]).

**Simulation with photons.** We now proceed to show how the monogamy violation can be implemented in an experimental demonstration.

Our experiment consists of a simulation of the OTC. The simulation consists in reconstructing all the statistics contained in the PDO $R_{123}$, which represents the OTC in our model, by constructing different sub-ensembles of entangled photon pairs, on which different measurements are realised. This experimental demonstration is therefore a proposal for a paradigm to realise a tomographic reconstruction of a PDO.

To this end, we generate a number of ensembles of entangled pairs of photons (A and B), each of which will be used to generate different statistics. Our setup is such that one photon (A) can be measured at two different times ($t_1$ and $t_2$), while the other one (B) can only be measured once at time $t_1$. In the simulation, the photon A measured at two different times represents the qubit entering the OTC and its copy emerging from the OTC, while the photon B represents the chronology-respecting qubit. Note that the simulation consists of reproducing the statistics of the OTC by performing the relevant measurement on different sub-ensembles—the quantum systems in each of these ensembles obey quantum theory and their quantum state is not a PDO.

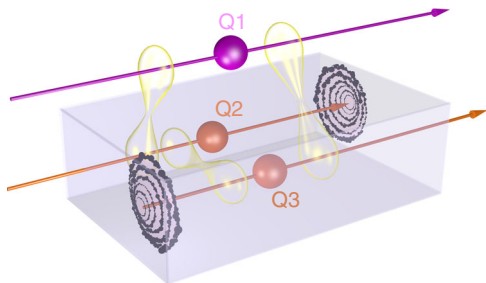

**Fig. 1** Open timelike curve circuit (pictorial representation). Qubits Q1 and Q2 are initially in a singlet state. Qubit Q2 enters a chronology-violating region, emerging as qubit Q3. In the chronology-violating region, qubits Q1 and Q2 must be in a singlet state, and so are qubits Q1 and Q3. Furthermore, since Q2 and Q3 are, respectively, the past and future copy of the same qubit, they are maximally correlated. This situation violates monogamy of entanglement: this is why it cannot be described by ordinary density operators, but it can be represented by PDOs

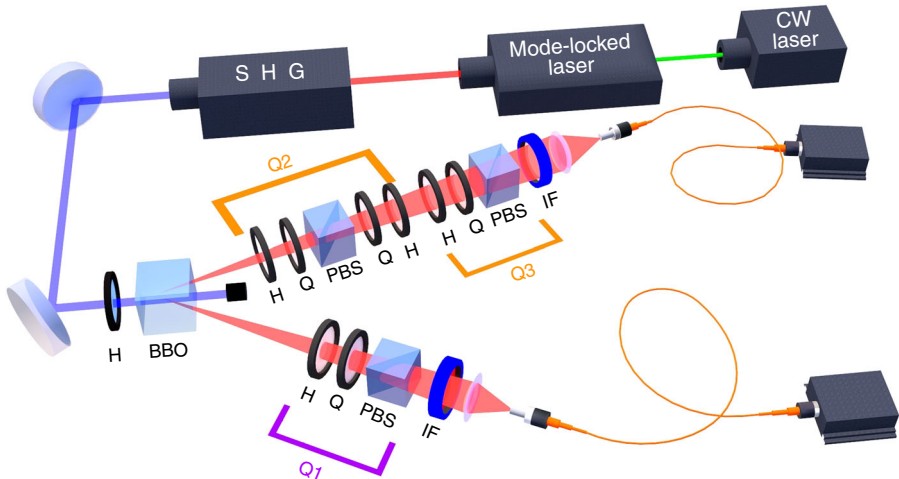

**Fig. 2** Experimental setup. A CW laser at 532 nm pumps a Ti:Sapphire crystal in an optical cavity, generating a mode-locked laser at 808 nm with a 76 MHz repetition rate. The pulsed laser is frequency doubled by second harmonic generation (SHG) and then injected into a 0.5-mm-thick $\beta$-barium borate (BBO) crystal, where degenerate non-collinear type-II parametric down-conversion (PDC) occurs. By spatially selecting the photons belonging to the intersections of the two PDC cones and properly compensating the temporal walk-off between the horizontal ($H$) and vertical ($V$) polarizations by adding a 0.25-mm-thick BBO crystal in both photon paths, we generate the entangled state $|\psi_-\rangle = \frac{1}{\sqrt{2}}(|HV\rangle - |VH\rangle)$. Afterwards, two polarisation measurements (Q2 and Q3) can be performed in sequence on branch A and one (Q1) on branch B. Correlations among them allow demonstrating violation of monogamy relation for PDO, simulating the scenario of OTC. H: half-wave plate; Q: quarter-wave plate; PBS: polarising beam splitter; IF: interference filter

In our setup (see Fig. 2), we exploit type-II parametric down-conversion (PDC) to generate the entangled state $|\psi_-\rangle = \frac{1}{\sqrt{2}}(|HV\rangle - |VH\rangle)$[26].

In order to evaluate both spatial and temporal correlations, in the photon A branch two polarisation measurements occur in cascade (Q2 and Q3), both carried by a half-wave plate (H) followed by a quarter-wave plate (Q) and a polarising beam splitter (PBS), while photon B branch hosts an identical H + Q + PBS unit (Q1). The quarter- and half-wave plates put after the PBS in Q2 counterbalance the polarisation rotation induced in the measurement process before the Q3 measurement takes place. The entangled photons are filtered by bandpass filters centred at $\lambda = 808$ nm (IFs, with 20 nm full-width at half-maximum (FWHM) on path A and 3 nm FWHM on path B) and coupled to multi-mode optical fibres connected to silicon single-photon avalanche diodes, whose outputs are sent to coincidence electronics.

To perform the reconstruction of the PDO $R_{123}$ we exploit different measurements to collect the three-point and the two-point correlations on the two photons. The three-point and two-point measurements are properly chosen in order to form a minimal quorum allowing for a full tomographic reconstruction[27] of $R_{123}$. This is needed because, in our experimental simulation, it would be impossible to perform a standard three-qubit quantum tomography procedure able to reconstruct $R_{123}$, since the measurement occurring on photon A at time $t_1$ (Q2) would obviously affect photon A at time $t_2$ (Q3) and the outcome of the measurement on it. To avoid this, we restrict ourselves to a particular sub-sample of the standard three-qubit tomographic measurements quorum in which the sequential measurement on photon A involves commuting observables, avoiding the issues derived from the measurement temporal ordering. The remaining information needed for the PDO reconstruction is obtained from the two-point correlation measurements.

In detail, for the two-point correlations this means preparing: (1) an ensemble where one measures, on photon A, the whole set of observables $\{X, Y, Z\}$ at time $t_1$ and the same set at time $t_2$, including all possible cross-correlations between different observables. This provides the full reconstruction of the reduced

pseudo-state $R_{23} = \frac{1}{4}(I + \Sigma_{23})$. (2) Another ensemble where one measures $X, Y, Z$ on photon A and on photon B at time $t_1$—this provides $R_{12}$. (3) A third ensemble where one measures $X, Y, Z$ on photon B at time $t_1$ and $X, Y, Z$ on photon A at time $t_2$. This provides $R_{13}$.

For the three-point correlations, this means preparing an ensemble where one measures $X, Y, Z$ on photon B and $X, Y, Z$ on photon A at time $t_1$, followed by measurements on photon A at time $t_2$ of the same observables measured on photon A at time $t_1$. From the conjectured $R_{123}$ we expect that the three-point correlations are all zero.

Our predictions are well confirmed by the simulation results. To the best of our knowledge, this result, shown in Fig. 3 compared to theoretical expectation, is the first tomographic reconstruction of a PDO.

This procedure highlights interesting properties of the PDO, which had gone unnoticed until now. Formally, just like for density operators, the reduced PDO of some subsystems is obtained by taking the trace on the degrees of freedom of the rest of the systems. For instance, in our case, $R_{13} = Tr_2(R_{123})$. However, unlike for density operators, $R_{13}$ cannot be reconstructed experimentally by using the measurements obtained for the three-point correlations and then averaging over the results of the measurements on the second qubit (i.e. photon A measured at time $t_1$). This is because the trace over a temporal degree of freedom is not equivalent to averaging with respect to all possible values of the observables that can be measured at that time. Indeed, $Tr(PR_{123})$, where $P$ is a generic projector could be negative, so that it cannot be interpreted in general as a probability (unless probabilities are allowed to take negative values[23]). This is a general property of PDOs. They are not always positive operators because the subsystems' degrees of freedom do not always represent spatial subsystems, but they could, instead, as in the case of qubits Q2 and Q3, represent timelike separated systems. The full tomographic reconstruction of a PDO is therefore different from reconstructing a standard density operator, as we have seen above.

All the reconstructions are in excellent agreement with the theoretical predictions, as certified by the fidelities obtained for

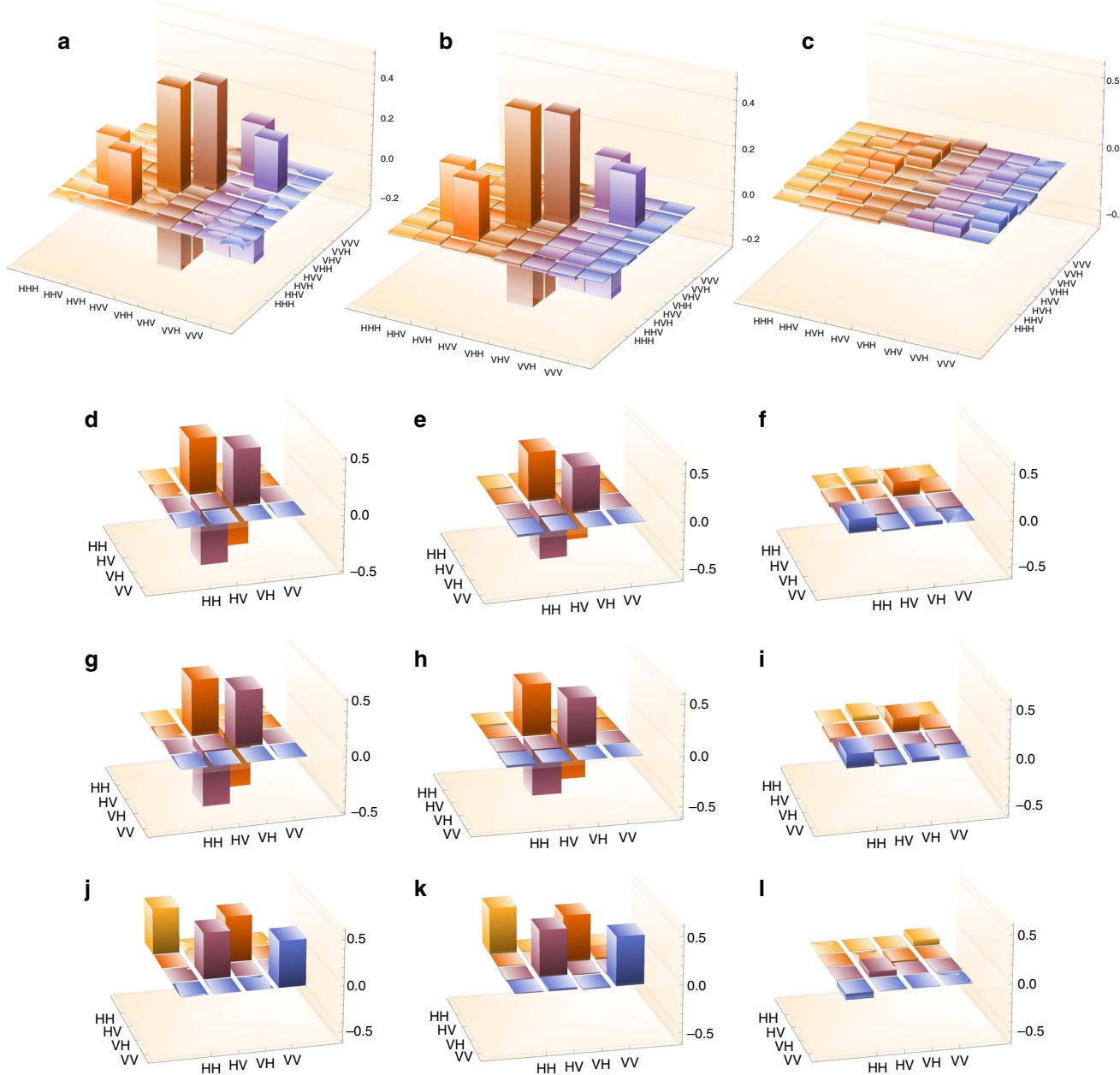

**Fig. 3** Pseudo-density operator tomographic reconstruction. Theoretical $R_{123}$ PDO (**a**: since $\mathrm{Im}[R_{123}] = 0$, we only plot $\mathrm{Re}[R_{123}]$) compared with the real (**b**) and imaginary (**c**) part extracted by quantum state tomography. Below, theoretical models of the $R_{12}$, $R_{13}$ and $R_{23}$ marginals (plots **d**, **g** and **j**, respectively) compared with the real (plots **e**, **h** and **k**) and imaginary (plots **f**, **i** and **l**) part of their tomographically reconstructed counterparts. Again, since in our model $\mathrm{Im}[R_{12}] = \mathrm{Im}[R_{13}] = \mathrm{Im}[R_{23}] = 0$, the corresponding theoretical plots have been omitted

the two 'physical' PDO marginals $R_{12}$ and $R_{13}$, that is, $F_{12} = 0.964$ and $F_{13} = 0.963$ (where $F_{ij}$ is the fidelity of the $R_{ij}$ density matrix with respect to the theoretically expected singlet state).

We also reconstruct the statistics from a CHSH test on the photon A at times $t_1$ and $t_2$, and on the photons A and B at time $t_1$, to show the predicted violation of monogamy. To this end, we evaluate the CHSH inequality on qubits Q2 and Q3, that is, on photon A at times $t_1$ and $t_2$ (temporal domain), obtaining the value $C_{23}^{(\mathrm{exp})} = 2.84 \pm 0.02$, in perfect agreement with the predicted violation. Then, we measure the CHSH on photons B and A at time $t_1$ (qubits Q1 and Q2, spatial domain), achieving $C_{12}^{(\mathrm{exp})} = 2.69 \pm 0.02$, a good violation of the classical bound. From these results, it follows:

$$C_{12}^{(\mathrm{exp})} + C_{23}^{(\mathrm{exp})} = 5.52 \pm 0.03 \tag{4}$$

showing a 160 standard deviations violation of the entanglement monogamy relation given by Eq. (3).

Furthermore, we extract the CHSH value related for the reconstructed $R_{13}$, obtaining $C_{13}^{(\mathrm{rec})} = 2.73$. This grants the remaining entanglement monogamy violations:

$$C_{12}^{(\mathrm{exp})} + C_{13}^{(\mathrm{rec})} = 5.42 \pm 0.07, \tag{5}$$

$$C_{23}^{(\mathrm{exp})} + C_{13}^{(\mathrm{rec})} = 5.55 \pm 0.07, \tag{6}$$

where, as uncertainty, we consider a 99% confidence interval on the experimental data.

The $C_{13}$ is extracted from the reconstructed PDO marginal $R_{13}$ because, in our simulation setup, a direct CHSH inequality

measurement for qubits 1 and 3 would be possible only leaving qubit 2 untouched, thus forbidding the possibility of measuring $C_{12}$.

## Discussion

Our proposal shows a radically different way of generalising quantum theory to describe chronology-violating regions containing an OTC, whose features we have simulated experimentally. $R_{123}$ is a viable descriptor of the physical situation where a qubit enters an OTC after having been entangled with another qubit. This is because, as we have demonstrated, it provides the same expected values for all the possible measurements that can be performed on those two qubits. It is a linear description in the sense that two different PDOs are related via a linear transformation. By proposing to use a PDO to describe the three qubits in the chronology-violating region, we depart from standard quantum mechanics, because we use a non-positive operator to describe the state of the qubits. Our proposal hints to a different way of formulating quantum theory, where, to describe a physical system with a certain dynamics, one gives the PDO as a faithful description of that physical situation. We implicitly define a PDO as faithful if it correctly describes the correlations between observables in different qubits. Now, once that step is taken, is it still possible to preserve some notion of linearity even when describing situations where properties like entanglement monogamy are violated? We conjecture that the answer is yes, because any two PDOs describing such different physical situations (e.g. two OTCs with different initial states) can be related by a linear transformation. This notion of linearity is, however, different from the linearity of quantum mechanical evolution. It would be interesting to understand the physical meaning of linear transformations between PDOs describing OTCs with different initial conditions, which we leave for a future paper. Also, a promising development of this proposal is a consistent general treatment of both OTCs and CTCs via PDOs. This could lead to a theory that retains linearity of quantum mechanics in a more general sense, but relaxes certain assumptions about the states of physical systems. Another interesting point is that, in the treatment of CTCs offered by[10], there is no violation of monogamy of entanglement. Extending this work to cover this type of CTCs is an interesting future step.

More generally, some models of quantum gravity might require spacetime to be quantised, whereby the distinction between timelike and spacelike degrees of freedom may become blurred below certain scales. This has prompted a number of proposals, for example, to modify the commutation relations of observables of different subsystems[28], or to incorporate indefinite causal order[1,29,30]. The pseudo-density formalism, in the light of what is proposed in this paper, might be a candidate to generalise the notion of quantum states to these scenarios.

## Data availability

The data that support the findings of this study are available from the corresponding author upon reasonable request.

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

## Acknowledgements

This research was supported by grant number (FQXi FFF Grant number FQXi-RFP-1812) from the Foundational Questions Institute and Fetzer Franklin Fund, a donor advised fund of Silicon Valley Community Foundation. CM's research was also supported by the Templeton World Charity Foundation and by the Eutopia Foundation. VV thanks the Oxford Martin School, the John Templeton Foundation, the EPSRC (UK). This work has received funding from the European Union's Horizon 2020 and the EMPIR Participating States in the context of the projects EMPIR-17FUN06 'SIQUST' and EMPIR-17FUN01 'BeCOMe'. This research is also supported by the National Research Foundation, Prime Minister's Office, Singapore, under its Competitive Research Programme (CRP Award No. NRF-CRP14-2014-02) and administered by Centre for Quantum Technologies, National University of Singapore.

## Author contributions

C.M. and V.V. (both responsible for the theoretical framework) proposed the experiment, and planned it together with F.P., A.A., I.P.D., M.Gram. and M.Gen. (responsible for the laboratories). The experimental realization was achieved (supervised by I.P.D., M.

Gram. and M.Gen.) by S.V., E.R., A.A. and F.P. The manuscript was prepared with inputs by all the authors, who also had a fruitful systematic discussion during the whole work development.

## Additional information

**Competing interests:** The authors declare no competing interests.

