## [Peer Review File · Nature Communications]

Reviewers' comments:

Reviewer #1 (Remarks to the Author):

The manuscript describes an application of the pseudo-density-matrix formalism to describe quantum systems near open timelike curves. The manuscript is well written and provides a good introduction to the problems associated with closed/open timelike curves. The results are interesting for a specialized audience, however, it is not clear what the significant contribution is to warrant publication in Nature Communications.

On the theory side, the manuscript is an application of the PDO formalism to a pair of entangled photons, where one is measured at two points in time. If one considers the two temporally separate measurements to represent measurements separate qubits connected through the OTC, one could presumably interpret the situation as a simulation of an OTC. However, the observed correlations, as one can see from the experimental implementation, are perfectly compatible with standard quantum evolution. It should be clarified how the treatment of the OTC scenario differs from conventional temporal quantum correlations.

In terms of the usefulness of the PDO approach, it is claimed in the abstract that, in contrast to other approaches, it is "preserving linearity and avoiding all other drastic consequences". This is in contradiction with the statement on page 3, that "nonlinearity is simulated by exploiting the effective nonlinearity from feeding extra information into the system". What nonlinearity and what extra information does this refer to? Moreover, there is no further discussion in the manuscript about how the PDO approach avoid the "drastic consequences" of other approaches. Instead the manuscript solely focuses on one particular aspect: the monogamy of entanglement, which is indeed violated in the PDO approach. There is also no further discussion on whether such a violation could lead to other undesirable consequences that might not be present in the usual treatments of OTCs.

The condition for entanglement monogamy in Eq.(3) is given without derivation or citation and not used further in the manuscript. Instead a different condition, namely monogamy of Bell-nonlocality in Eq.(4) is used for the rest of the manuscript. A violation of this condition for temporal correlations is not surprising and was predicted in [arxiv: quant-ph/0402127]. There is no discussion of the relevance of this violation in the presence or absence of an OTC.

The experiment itself is a simple two-photon experiment with one intermediate projective measurement, combined with state tomography. Similar experiments have been used for many years in tests of Leggett-Garg inequalities and spatio-temporal quantum correlations. It is also not surprising that one can perform tomography of the PDO, although the manuscript points out a few subtleties, yet does not give a complete description of the reconstruction procedure.

With regards to the difficulties of reconstruction discussed on page 4, it has been argued in [Proc. R. Soc. A 473 (2205)] that approaches with one Hilbert space per time-step are limited. It would be interesting to explore how double-Hilbert space approaches, such the process matrix framework might handle these.

Finally, it is not explained why the value of C_{13} is only extracted from tomography, rather than measured directly. This is particularly important, since the measured monogamy violation in Eq. (5) is trivial, given that C_{23} is measured on the same qubit and this always shows perfect correlations. In contrast the monogamy of C_{12} and C_{13} involves measurements of different qubits. This also raises the question how these correlations would be measured in practice, since it involves a measurement of the two versions of the same qubit existing in the chronology violating region. One would naively expect that the measurement of one of them should have a causal effect on the other, since they are connected via the OTC. Some more discussion would be appropriate.

Reviewer #2 (Remarks to the Author):

I have read 'Quantum Entanglement Near Open Timelike Curves: Theory and Experiment' by Marletto et al., which was submitted to Nature Communications. This paper applies a theoretical construction---the pseudo-density operator, designed to combine temporal and spatial correlations on a similar footing---to the problem of describing quantum systems in spacetimes with closed timelike curves (CTCs). This problem has attracted an increasing amount of interest, with several proposals having been studied recently: in particular, the approach of Deutsch (D-CTCs), and the postselected CTC model of Bennett and Schumacher (P-CTCs). However, in both cases, the quantum evolution in general becomes nonlinear. This is particularly the case for D-CTCs, where passing a system through a CTC breaks entanglement with any chronology-respecting system, even if the chronology-violating system does not interact with its past self (what Ralph et al. called an 'open timelike curve' or OTC).

The pseudo-density operator (PDO) is an operator that acts on tensor products of Hilbert spaces in such a way that it reproduces the correlations predicted by quantum mechanics. But unlike a usual density operator, the different tensor product 'slots' in the operator can represent not only spatially separated subsystems, but the same subsystem at different times. In order to correctly reproduce the results of quantum mechanics, this operator is not positive in general, but its properties otherwise resemble those of a density operator (i.e., it is Hermitian and has trace 1). Most of the calculations in this paper consider the (fairly) simple case of a PDO with three qubit 'slots,' two for the same qubit A at two different times, and the third for a spatially separated qubit B. The authors apply this description to an OTC, and show if qubits A and B are maximally entangled before the CTC forms, then they can retain entanglement after qubit A has passed back through the CTC.

The authors also did an experimental simulation of this process in a quantum optical implementation, using spontaneous parametric downconversion to produce pairs of maximally-entangled measurement, and doing a modified quantum state tomography protocol (involving measurements at multiple places and times) to reconstruct the PDO. This process is interesting, though the agreement of the experiment with theory is hardly surprising, since that is exactly what it was designed to do.

Indeed, in the absence of an actual CTC to experiment with, I think the authors could more valuably have spent the paper discussing the theory in more detail, rather than the experimental simulation. But I suppose the modified quantum state tomography procedure is an interesting thing to demonstrate experimentally, even apart from the simulations of CTCs.

In general, I found this paper interesting and well-written. The idea of using PDOs to describe CTCs and other systems with unusual causality structures is an interesting one, and I would be very interested to see if a fully consistent theory could be developed that goes beyond OTCs. In particular, in the case where a particle interacts with its past self, could one find a theory that retains the linearity of quantum mechanics? I am a bit doubtful, since every approach proposed so far has exhibited nonlinearity to a greater or lesser degree. But it would be interesting to find out.

I do have some comments about the discussion of prior work on quantum mechanics with CTCs, as well as the results presented. First, one of the most important results in this earlier work is missing: the paper of Aaronson and Watrous (Proc. Roy. Soc. A 465, 631 (2009)) showing that Deutsch CTCs (D-CTCs) (and their classical analogues) allow problems in PSPACE to be solved efficiently. Also, while the authors cite references [8] and [10] as showing that D-CTCs allow cloning, this was actually shown to be true in general by Brun, Wilde and Winter (PRL 111, 190401 (2013)).

Somewhat more seriously, the discussion and references in this paper conflate D-CTCs and P-CTCs. Most of the work cited in the references is on D-CTCs, which is also what the discussion in the paper assumes (without really acknowledging that there are at least two different approaches). But reference [4] by Lloyd et al. is about P-CTCs (and includes an experimental simulation of them that is in many ways more interesting than the one cited in [6]). The properties of D-CTCs and P-CTCs are quite different. While computers with P-CTCs are much more powerful than ordinary quantum computers, they are significantly less powerful than computers with D-CTCs. Like D-CTCs, P-CTCs allow nonorthogonal states to be distinguished, but only a linearly independent set of them. And unlike D-CTCs, P-CTCs do not always have valid solutions; there are a set of singular points corresponding to events with probability zero (grandfather paradoxes).

In the context of the current paper, one major difference is important: P-CTCs are not necessarily entanglement breaking (unlike D-CTCs), and in particular their predictions for CTCs are exactly the same as those in this paper. I would not be at all surprised if any consistent theory of CTCs based on PDOs turns out to be essentially equivalent to P-CTCs: a linear but non-unitary evolution followed (in general) by renormalization.

PDOs are a very interesting construction, and I strongly suspect that they are closely related to the work from the 1990s of Chris Isham on consistent/decoherence histories. In this work, Isham exploited the following mathematical trick relating an ordinary product of operators to a tensor product:

$$\text{Tr}\{ ABC \} = \text{Tr}\{ S(A \otimes B \otimes C) \}$$

where S is the cyclic shift operator. (This generalizes to any number of operators.) Using this trick, Isham was able to replace the 'history operators' of consistent histories, which were products of projectors, with tensor products of projectors (which are therefore projectors themselves). To do this, he replaced the density matrix at the starting time by a product of the density matrix with shift operators. This product seems to play essentially the same role as a PDO in the current work. I suggest that the authors might find some interesting connections if they look into this work.

On the whole I liked this paper and found it interesting. With revisions based on the comments above, I believe it should be publishable. I am not quite so firm on whether it should be published in Nature Communications rather than elsewhere; however, I lean in the positive direction. Unless other referees have spotted more serious problems that I missed, I would recommend this paper for publication in Nature Communications.

Reviewer #3 (Remarks to the Author):

The authors present a new result about a violation of standard quantum mechanical rules (the monogamy of entanglement) in a situation where a system goes back in time. Unlike previous discussions on this topic, there are no explicit closed time-like curves in the situation. They then show an experiment that reproduces the predicted violation of a Bell-like inequality, without time-travel.

The result is new and certainly interesting. It can be interpreted in various ways (that do not necessarily agree with the authors' view), and it would almost certainly generate interesting discussions in the community. The paper is also very well written for the most part, and is accessible to a very wide audience. I recommend publication, however, there are issues that should be addressed before publication (probably in supplementary material), see details below.

1. The theory seems to be correct, however there are not enough details so it is difficult to verify

that the results are correct. The mathematical formalism is relatively new and there should be more details explaining how the PDOs are constructed, and how PDO tomography should work.

2. Since the PDOs have negative entries, it is clear that they predict negative 'probabilities' for certain measurements. This does not seem to cause any issues, probably for good reasons (I assume that it is impossible to make these types of measurements), but that should be explained, possibly with an example.

3. The experimental protocol is not clear. In particular since only two detectors are used, it seems like there is some type of post-selection (what happens to photons that exit from the other side of the PBS?). How is this justified?

4. The authors write "Our proposal shows a way of generalising quantum theory to describe chronology-violating regions, whose features we have demonstrated experimentally". The second part of this sentence is misleading, the experiment does not involve the chronology-violating regions and is linear in time, it is more of an analogy.

Reply to referees

We thank the Referees for their sharp criticism and useful comments, allowing us to improve the quality of our manuscript. In the following, we provide a point-by-point reply to Their reports. (Referees' comments are in boldface, our replies in plain text).

Reviewer #1 (Remarks to the Author):

The manuscript describes an application of the pseudo-density-matrix formalism to describe quantum systems near open timelike curves. The manuscript is well written and provides a good introduction to the problems associated with closed/open timelike curves.

We thank the Referee for Her/His appreciation and comments.

The results are interesting for a specialized audience, however, it is not clear what the significant contribution is to warrant publication in Nature Communications.

The paper contains two main points that are of interest to a wide audience such as Nature Communication's: 1) it is a theoretical proposal for a novel description of qubits in Open Time-like curves with pseudo-density operators (PDOs). This represents a major novelty, because there are at present only 'reduced state' descriptions for qubits in open *and* closed timelike curves (given by the relevant consistency condition). The difficulty in describing the overall state with density operators is that the system consisting of the past and future qubits is *not* equivalent to two spatially separated qubits identically prepared. This can be seen by considering what happens by applying some operation affecting the past qubit, which affects the future qubit; or by entangling the qubit in the chronology-respecting region with another, chronology respecting qubit: the density operator fails to describe the composite system in this case because monogamy of entanglement is violated. The PDO, instead, allows one to describe the latter scenario, as we explain in our paper. 2) The other significant contribution of the paper is to propose *a new type of tomographic reconstruction*, which can be reinterpreted as a simulation of multi-time PDOs.

In addition, the study of space-time correlation is ubiquitous in fundamental physics, and at the heart of one of the clashes between quantum theory (where space and time are treated as different entities); and relativity, where they are part of a single dynamical entity. Therefore, the topics touched in this paper are not just relevant for open time-like curves, but for a number of other open issues in fundamental physics. These two points for instance are central to current attempts to extend the quantum descriptions to systems that do not obey quantum theory strictly, but go beyond it by violating its characteristic properties.

As such, this work is likely to impact the studies of descriptions to incorporate into quantum theory the possibility of indefinite causal order (see e. g. ref. [1]), quantum gravity, and even black holes. Thus, this work nicely complements existing contributions in Nature Communication on the same topic, e.g. <https://www.nature.com/articles/ncomms5145#f1>, by proposing a novel experimental and theoretical study of OTCs, with possible applications to the study of space-time correlation in several other contexts.

We have rephrased part of the introduction to illustrate these points:

“Quantum theory and general relativity each provide well-verified predictions, in their respective domains. However, they also provide predictions that cannot yet be probed experimentally but give one the opportunity of exploring physics which is rather different from what we perceive directly at our scales. Of particular interest are predictions of space-time correlations violating the standard properties of quantum theory, such as superpositions of different space-time geometries in quantum gravity, resulting in superposing different causal orders; or the physics of black holes. In these cases, it is possible to relax some of the assumptions of quantum theory and still have a coherent picture - which leads to proposals for new frameworks that go beyond quantum theory. An important example of such violations is the dynamics of a quantum system near closed timelike curves (CTC).”

And also:

“Our proposal consists of two parts. First, [...]. The second part of our proposal is an experimental demonstration... presents interesting subtleties.”

On the theory side, the manuscript is an application of the PDO formalism to a pair of entangled photons, where one is measured at two points in time.

This is not the case. This is a point that requires clarification. Our theoretical analysis is a proposed model (relying on a PDO) describing a qubit entering an OTC, which was entangled with another chronology-respecting qubit in the distant past. The fact that this physical situation is then SIMULATED with a pair of entangled photons measured at different times is a different matter. The PDO we propose is not a quantum state and cannot represent the state of two entangled qubits – indeed, it is a three-qubit operator, not a two-qubit operator. The experimentally reconstructed statistics are the same as those described by the PDO we propose, but they are NOT the PDO (the PDO is an observable of a 3-qubit system, not of the two-qubit system of the experiment). We have added a sentence to clarify this point:

“The simulation of the OTC consists in reproducing the correlations in the PDO we conjecture can describe the OTC.”

If one considers the two temporally separate measurements to represent measurements separate qubits connected through the OTC, one could presumably interpret the situation as a simulation of an OTC.

Yes, the Reviewer is right.

However, the observed correlations, as one can see from the experimental implementation, are perfectly compatible with standard quantum evolution.

Actually, this is not the case. The observed correlations are the result of averages on different sub-ensembles, which cannot be described by any quantum state/quantum evolution. As we said, the experimental part of this paper is only a SIMULATION of the OTC (as one would expect). We have added a sentence to explain this. We now write:

“Our experiment consists of a simulation of the OTC. The simulation consists in reconstructing all the statistics contained in the PDO R_{123} , which represents the OTC in our

model, by constructing different sub-ensembles of entangled photon pairs, on which different measurements are realised.”.

It should be clarified how the treatment of the OTC scenario differs from conventional temporal quantum correlations.

In the OTC scenario there are a past qubit, a future qubit, and a chronology-respecting qubit. The past and future qubits are two distinct entities, which are nonetheless time-like separated, because of the OTC. The fact that they are distinct is shown by the fact that, in the OTC model, each one is separately maximally entangled with a third qubit, at a given time. Standard time quantum correlations describe, on the other hand, the SAME qubit considered at different times. Our model for the OTC suggests that a linear operator like the PDO, which can be used to represent quantum time correlations on the SAME qubits at two different times, can be also used to represent the past and future copies of a qubit in an OTC – this is one of the novel elements of our proposal.

We have added this comment to explain all this (last line of page 2 and onwards):

“Note that this is a novel application of the PDO to describe two distinct time-like separated qubits, i.e. the past and future copy of the qubit within the OTC, which are perfectly correlated in all bases. This is different from the standard use of the PDO as a tool to describe time-like correlations (which are already known to violate monogamy of entanglement when considering the time-evolution of a single qubit [24]).”.

In terms of the usefulness of the PDO approach, it is claimed in the abstract that, in contrast to other approaches, it is “preserving linearity and avoiding all other drastic consequences”.

The Reviewer is right – this requires additional explanation. The PDO is a linear operator that can describe channels which cause violation of monogamy (and therefore violate quantum theory). However, this description preserves linearity in the sense that two different PDOs describing OTCs with different initial conditions are related by a linear transformation. We have changed the statement in the abstract to this more accurate statement: “Here we propose an alternative approach, maintaining that monogamy of entanglement is violated.” And we have added another sentence in the text:

“As we shall see, our approach allows one to preserve linearity in an interesting way - because any two different PDOs are related by a linear transformation. This opens a new line of investigation where instead of modifying the linearity of quantum theory we modify other features, specifically the positivity of the quantum state, to accommodate features induced by other physical requirements, in this case general relativity.”.

And also:

“Interestingly, as we said, this can be thought of as preserving linearity because any two PDOs (of the same dimensionality) are hermitian operators which can be related to one another via a linear transformation.”.

This is in contradiction with the statement on page 3, that “nonlinearity is simulated by exploiting the effective nonlinearity from feeding extra information into the system”. What nonlinearity and what extra information does this refer to?

We have modified this sentence too, which was anyway referring to a different experiment (ref. [8]). Now we write (page 3, right column):

“In the simulation, the photon A measured at two different times represents the qubit entering the OTC and its copy emerging from the OTC; while the photon B represents the chronology-respecting qubit.”.

Moreover, there is no further discussion in the manuscript about how the PDO approach avoid the “drastic consequences” of other approaches. Instead the manuscript solely focuses on one particular aspect: the monogamy of entanglement, which is indeed violated in the PDO approach. There is also no further discussion on whether such a violation could lead to other undesirable consequences that might not be present in the usual treatments of OTCs.

Good point. We have added a paragraph to explain that this is an open problem – one of the possible interesting future developments of this work:

“It is a linear description in the sense that two different PDOs are related via a linear transformation. It would be interesting to understand the physical meaning of linear transformations between PDOs describing OTCs with different initial conditions, which we leave for a future paper. Also, a promising development of this proposal is a consistent general treatment of both OTCs and CTCs via PDOs. This could lead to a theory that retains linearity of quantum mechanics in a more general sense, but relaxes certain assumptions about the states of physical systems.”.

The interesting thing is that for the OTC case we can still use the PDO, a linear operator, to describe the state of the three qubits in the chronology violating region. We also conjecture that once an interaction is added, in the case of the CTCs, the PDO is still a good candidate for a description – but this is a conjecture that we leave open for future work.

The condition for entanglement monogamy in Eq.(3) is given without derivation or citation and not used further in the manuscript.

Yes, true. We have remedied to that by removing the equation and using directly the Bell inequality criterion.

Instead a different condition, namely monogamy of Bell-nonlocality in Eq.(4) is used for the rest of the manuscript. A violation of this condition for temporal correlations is not surprising and was predicted in [arxiv: quant-ph/0402127]. There is no discussion of the relevance of this violation in the presence or absence of an OTC.

The point raised by the Reviewer is relevant, and we have added a reference to that publication and a comment to explain how they are related, writing:

“Note that this is a novel application of the PDO to describe two distinct time-like separated qubits, i.e. the past and future copy of the qubit within the OTC, which are perfectly correlated in all bases. This is different from the standard use of the PDO as a tool to describe time-like correlations (which are already known to violate monogamy of entanglement when considering the time-evolution of a single qubit [24]).”.

The experiment itself is a simple two-photon experiment with one intermediate projective measurement, combined with state tomography. Similar experiments have been used for many years in tests of Leggett-Garg inequalities and spatio-temporal quantum correlations. It is also not surprising that one can perform tomography of the PDO, although the manuscript points out a few subtleties, yet does not give a complete description of the reconstruction procedure.

The key point of the experimental part of this paper is to set a paradigm to construct a three-point PDO (generalizable to n-point). So, even though the basis of the experiment is a pair of entangled photons, the innovative and interesting part is how to realise the tomography of a PDO– which, as we explain in the paper, is different from the tomography of a 3-qubit density operator. This is because, although the operator in question is a 3-qubit operator, the slots of the tensor-product are not space-like separated qubits. We have now improved the description of the reconstruction, and explained why it is a major new improvement on the existing other tomographies of PDOs existing in the literature. We agree that it is not surprising that one can do the reconstruction of the PDO, on the other hand almost no experiment of quantum Information, as usual tomography, are surprising once a theoretical prediction is made: they just show the practical feasibility of theoretical methods, that can then be exploited in quantum technologies. As in many other experiments based on PDC entangled photons (e.g. the ones on Leggett-Garg inequalities) the set-up is rather standard, what changes is the way it is exploited for and, therefore, the details of the apparatus.

We have improved on the presentation of the experimental part by adding this paragraph:

“Our experiment consists of a simulation of the OTC. The simulation consists in reconstructing [...] by performing the relevant measurement on different subensembles - the quantum systems in each of these ensembles obey quantum theory and their quantum state is not a PDO.”

With regards to the difficulties of reconstruction discussed on page 4, it has been argued in [Proc. R. Soc. A 473 (2205)] that approaches with one Hilbert space per time-step are limited. It would be interesting to explore how double-Hilbert space approaches, such the process matrix framework might handle these.

These are not difficulties, but interesting features of the PDO approach, which also reflect on the interpretation of the PDO itself. As suggested, we have added a sentence (page 2, left column) observing how it would be interesting to extend this work to analyse other approaches under this scenario:

“An interesting application of this work would be to consider how other approaches to incorporate space-time correlations in quantum theory [17] could be used to the same effect as the PDO in this context.”

Finally, it is not explained why the value of C_{13} is only extracted from tomography, rather than measured directly. This is particularly important, since the measured monogamy violation in Eq. (5) is trivial, given that C_{23} is measured on the same qubit and this always shows perfect correlations. In contrast the monogamy of C_{12} and C_{13} involves measurements of different qubits.

Eq. (5) where we compute, rather than measure directly, C_{13} was inserted for convenience of presentation and also to show consistency in the experimental procedure. C_{13} however corresponds to measuring the Bell inequality on photon A at time t_1 and on photon B at time t_2 , without acting on photon B at time t_1 . This means that, in our simulation, to obtain a direct C_{13} measurement we should leave qubit 2 untouched, forbidding us to obtain C_{12} and C_{23} . Finally, of course C_{23} is somehow trivial, but this is unavoidable in such kind of simulations (as was the case in ref. [8] as well).

This also raises the question how these correlations would be measured in practice, since it involves a measurement of the two versions of the same qubit existing in the chronology violating region.

Indeed, to avoid this issue, we chose to extract C_{13} from the reconstructed PDO because in our specific set-up would be impossible to perform the measurement 1-3 without a previous measurement 1-2. This is now explicitly declared in the sentence (page 6, left column):

“The C_{13} is extracted from the reconstructed PDO marginal R_{13} because, in our simulation setup, a direct CHSH inequality measurement for qubits 1 and 3 would be possible only leaving qubit 2 untouched, thus forbidding the possibility of measuring C_{12} and, as a consequence, the observation of the first entanglement monogamy violation reported in Eq. (5).”

One would naively expect that the measurement of one of them should have a causal effect on the other, since they are connected via the OTC. Some more discussion would be appropriate.

Yes indeed – this is a very interesting point, which is why the notion of ‘monogamy violation’ for OTC’s, as presented in the literature, seems to us perhaps not completely adequate. Although the entanglement properties are violated in the chronology violating region, verifying the violation or using that entanglement for other purposes would require to act on the qubit entering the open time-like curve, which would affect the state of the qubit. We have a footnote to warn the reader about this:

“Although monogamy of entanglement is violated in the chronology violating region, verifying the violation seems practically hard, because it would require to act on the qubit entering the open time-like curve, which would affect the state of the qubit itself.”

Reviewer #2 (Remarks to the Author):

I have read 'Quantum Entanglement Near Open Timelike Curves: Theory and Experiment' by Marletto et al., which was submitted to Nature Communications. This paper applies a theoretical construction---the pseudo-density operator, designed to combine temporal and spatial correlations on a similar footing---to the problem of describing quantum systems in spacetimes with closed timelike curves (CTCs). This problem has attracted an increasing amount of interest, with several proposals having been studied recently: in particular, the approach of Deutsch (D-CTCs), and the postselected CTC model of Bennett and Schumacher (P-CTCs). However, in both cases, the quantum evolution in general becomes nonlinear. This is particularly the case for D-CTCs, where passing a system through a CTC breaks entanglement with any chronology-respecting system, even if the chronology-violating system does not interact with its past self (what Ralph et al. called an 'open timelike curve' or OTC).

The pseudo-density operator (PDO) is an operator that acts on tensor products of Hilbert spaces in such a way that it reproduces the correlations predicted by quantum mechanics. But unlike a usual density operator, the different tensor product 'slots' in the operator can represent not only spatially separated subsystems, but the same subsystem at different times. In order to correctly reproduce the results of quantum mechanics, this operator is not positive in general, but its properties otherwise resemble those of a density operator (i.e., it is Hermitian and has trace 1). Most of the calculations in this paper consider the (fairly) simple case of a PDO with three qubit 'slots,' two for the same qubit A at two different times, and the third for a spatially separated qubit B. The authors apply this description to an OTC, and show if qubits A and B are maximally entangled before the CTC forms, then they can retain entanglement after qubit A has passed back through the CTC.

The authors also did an experimental simulation of this process in a quantum optical implementation, using spontaneous parametric downconversion to produce pairs of maximally-entangled measurement, and doing a modified quantum state tomography protocol (involving measurements at multiple places and times) to reconstruct the PDO. This process is interesting,

We thank the Referee for the remarks and the succinct exposition of our paper.

though the agreement of the experiment with theory is hardly surprising, since that is exactly what it was designed to do.

Yes. This is a simulation. It is designed to be so. However, the interesting part in the experiment is the procedure to reconstruct a 3-qubit PDO (see the reply to referee 1). We have added a sentence to explain that (page 2, left column):

"The second part of our proposal is an experimental demonstration of the statistics of the OTC, where we simulate the entanglement monogamy violation and provide a full tomographic reconstruction of the whole PDO. This sets the paradigm for the experimental reconstruction of the PDO, which as we shall explain presents interesting subtleties."

Indeed, in the absence of an actual CTC to experiment with, I think the authors could

more valuably have spent the paper discussing the theory in more detail, rather than the experimental simulation. But I suppose the modified quantum state tomography procedure is an interesting thing to demonstrate experimentally, even apart from the simulations of CTCs.

Yes indeed, as explained above. We have added a comment to highlight this fact (see also the previous reply):

“Our proposal consists of two parts. First, [...]. The second part of our proposal is an experimental demonstration... presents interesting subtleties.”

In general, I found this paper interesting and well-written. The idea of using PDOs to describe CTCs and other systems with unusual causality structures is an interesting one, and I would be very interested to see if a fully consistent theory could be developed that goes beyond OTCs. In particular, in the case where a particle interacts with its past self, could one find a theory that retains the linearity of quantum mechanics? I am a bit doubtful, since every approach proposed so far has exhibited nonlinearity to a greater or lesser degree. But it would be interesting to find out.

We thank the Referee for the remark. We have added some discussion to highlight how this is a possible application of our work (page 6, left column):

“Our proposal shows a novel way of generalising quantum theory to describe chronology-violating regions containing an OTC, whose features we have simulated experimentally. R_{123} is a viable descriptor of the physical situation where a qubit enters an OTC after having been entangled with another qubit. This is because, as we have demonstrated, it provides the same expected values for all the possible measurements that can be performed on those two qubits. It is a linear description in the sense that two different PDOs are related via a linear transformation. It would be interesting to understand the physical meaning of linear transformations between PDOs describing OTCs with different initial conditions, which we leave for a future paper. Also, a promising development of this proposal is a consistent general treatment of both OTCs and CTCs via PDOs. This could lead to a theory that retains linearity of quantum mechanics in a more general sense, but relaxes certain assumptions about the states of physical systems. Another interesting point is that in the treatment of CTCs offered by [9], there is no violation of monogamy of entanglement. Extending this work to cover this type of CTCs is an interesting future step.”

I do have some comments about the discussion of prior work on quantum mechanics with CTCs, as well as the results presented. First, one of the most important results in this earlier work is missing: the paper of Aaronson and Watrous (Proc. Roy. Soc. A 465, 631 (2009)) showing that Deutsch CTCs (D-CTCs) (and their classical analogues) allow problems in PSPACE to be solved efficiently. Also, while the authors cite references [8] and [10] as showing that D-CTCs allow cloning, this was actually shown to be true in general by Brun, Wilde and Winter (PRL 111, 190401 (2013)).

We thank the Referee for the comment. We have added those references.

Somewhat more seriously, the discussion and references in this paper conflate D-CTCs and P-CTCs. Most of the work cited in the references is on D-CTCs, which is also what the discussion in the paper assumes (without really acknowledging that there are at least two different approaches). But reference [4] by Lloyd et al. is about P-CTCs (and includes an experimental simulation of them that is in many ways more interesting than the one cited in [6]). The properties of D-CTCs and P-CTCs are quite different. While computers with P-CTCs are much more powerful than ordinary quantum computers, they are significantly less powerful than computers with D-CTCs. Like D-CTCs, P-CTCs allow nonorthogonal states to be distinguished, but only a linearly independent set of them. And unlike D-CTCs, P-CTCs do not always have valid solutions; there are a set of singular points corresponding to events with probability zero (grandfather paradoxes).

In the context of the current paper, one major difference is important: P-CTCs are not necessarily entanglement breaking (unlike D-CTCs), and in particular their predictions for OTCs are exactly the same as those in this paper. I would not be at all surprised if any consistent theory of CTCs based on PDOs turns out to be essentially equivalent to P-CTCs: a linear but non-unitary evolution followed (in general) by renormalization.

This is an interesting comment, and we have added a paragraph to alert the reader about this possibility:

“Another interesting point is that in the treatment offered by [9], there is no violation of monogamy of entanglement.”

PDOs are a very interesting construction, and I strongly suspect that they are closely related to the work from the 1990s of Chris Isham on consistent/decoherence histories. In this work, Isham exploited the following mathematical trick relating an ordinary product of operators to a tensor product:

$$\text{Tr}\{ABC\} = \text{Tr}\{S(A \otimes B \otimes C)\}$$

where S is the cyclic shift operator. (This generalizes to any number of operators.) Using this trick, Isham was able to replace the ‘‘history operators’’ of consistent histories, which were products of projectors, with tensor products of projectors (which are therefore projectors themselves). To do this, he replaced the density matrix at the starting time by a product of the density matrix with shift operators. This product seems to play essentially the same role as a PDO in the current work. I suggest that the authors might find some interesting connections if they look into this work.

Yes, this is true. We have added a reference to it also to explain that the context in which Isham introduced this idea is completely different from that of PDOs (page 2, right column):

“We note that a similar formal tool was already introduced by [20] in the context of the consistent-history approach”.

On the whole I liked this paper and found it interesting. With revisions based on the

comments above, I believe it should be publishable. I am not quite so firm on whether it should be published in Nature Communications rather than elsewhere; however, I lean in the positive direction.

We thank the Referee for Her/His appreciation and positive evaluation of our work.

Unless other referees have spotted more serious problems that I missed, I would recommend this paper for publication in Nature Communications.

Reviewer #3 (Remarks to the Author):

The authors present a new result about a violation of standard quantum mechanical rules (the monogamy of entanglement) in a situation where a system goes back in time. Unlike previous discussions on this topic, there are no explicit closed time-like curves in the situation. They then show an experiment that reproduces the predicted violation of a Bell-like inequality, without time-travel.

The result is new and certainly interesting. It can be interpreted in various ways (that do not necessarily agree with the authors' view), and it would almost certainly generate interesting discussions in the community. The paper is also very well written for the most part, and is accessible to a very wide audience.

Let us thank the Referee for Her/His positive comment on our paper.

I recommend publication, however, there are issues that should be addressed before publication (probably in supplementary material), see details below.

1. The theory seems to be correct, however there are not enough details so it is difficult to verify that the results are correct. The mathematical formalism is relatively new and there should be more details explaining how the PDOs are constructed, and how PDO tomography should work.

Yes. As explained in the replies to the other Referees, we have added a more careful description of the process of tomography and also a discussion of the theoretical subtleties. First, we have added a reference to the work by Horsman et al. (ref. [16]) on the formal properties of the PDO. Also, in the experimental section we now write:

“Our experiment consists of a simulation of the OTC. The simulation consists in reconstructing all the statistics contained in the PDO R_{123} , which represents the OTC in our model, by constructing different sub-ensembles of entangled photon pairs, on which different measurements are realised. This experimental demonstration is therefore a proposal for a paradigm to realise a tomographic reconstruction of a PDO.

To this end, we generate a number of ensembles of entangled pair of photons (A and B), each of which will be used to generate different statistics. Our setup is such that photon (A) can be measured at two different times (t_1 and t_2) while the other one (B) can only be measured once at

time t_1 . In the simulation, the photon A measured at two different times represents the qubit entering the OTC and its copy emerging from the OTC; while the photon B represents the chronology-respecting qubit. Note that the simulation consists of reproducing the statistics of the OTC by performing the relevant measurement on different sub-ensembles - the quantum systems in each of these ensembles obey quantum theory and their quantum state is not a PDO.”.

And also:

“[...] This is needed because, in our experimental simulation, it would be impossible to perform a standard 3-qubit quantum tomography procedure able to reconstruct R_{123} , since the measurement occurring on photon A at time t_1 (qubit 2) would obviously affect photon A at time t_2 (qubit 3) and the outcome of the measurement on it. To avoid this, we restrict ourselves to a particular sub-sample of the standard 3-qubit tomographic measurements quorum in which the sequential measurement on photon A involves commuting observables, avoiding the issues derived from the measurement temporal ordering. The remaining information needed for the PDO reconstruction is obtained from the 2-point correlation measurements. In detail, [...]. For the 3-point correlations, this means preparing an ensemble where one measures X, Y, Z on photon B and X, Y, Z on photon A at time t_1 , followed by measurements on photon A at time t_2 of the same observables measured on photon A at time t_1 .”.

2. Since the PDOs have negative entries, it is clear that they predict negative ‘probabilities’ for certain measurements. This does not seem to cause any issues, probably for good reasons (I assume that it is impossible to make these types of measurements), but that should be explained, possibly with an example.

We have also added reference 21 and a comment to explain how the example we had included, about a qubit measured at two different times, clarifies what these ‘negative probabilities’ (negative expected values of projectors) are and what they mean physically:

“This means that it presents negative expectation values of projectors. For example, R12 has the singlet state as an eigenstate, with eigenvalue $-1/2$ [21]. This is interpreted as the signature of correlations in time [22].”.

3. The experimental protocol is not clear. In particular, since only two detectors are used, it seems like there is some type of post-selection (what happens to photons that exit from the other side of the PBS?). How is this justified?

Yes, the Referee is right: we have some postselection. This corresponds to selecting a specific state in the projective measurement and it is the usual method for performing tomographic reconstructions. In fact, all the measurements involved are projections onto selected polarizations (with the PBSs behaving like polarizers), forming a tomographic set. We have now expanded on the explanation of the experiment:

“To this end, we generate a number of ensembles of entangled pair of photons (A and B), each of which will be used to generate different statistics. Our setup is such that photon (A) can be measured at two different times (t_1 and t_2) while the other one (B) can only be measured once at time t_1 . In the simulation, the photon A measured at two different times represents the qubit

entering the OTC and its copy emerging from the OTC; while the photon B represents the chronology-respecting qubit. Note that the simulation consists of reproducing the statistics of the OTC by performing the relevant measurement on different subensembles - the quantum systems in each of these ensembles obey quantum theory and their quantum state is not a PDO."

4. The authors write "Our proposal shows a way of generalising quantum theory to describe chronology-violating regions, whose features we have demonstrated experimentally". The second part of this sentence is misleading, the experiment does not involve the chronology-violating regions and is linear in time, it is more of an analogy.

True. We have corrected the sentence to say that what we propose is a *simulation* of the features of our proposed model. We now write:

"Our proposal shows a novel way of generalising quantum theory to describe chronology-violating regions containing an OTC, whose features we have simulated experimentally."

Reviewers' comments:

Reviewer #1 (Remarks to the Author):

The manuscript has improved from the previous version and could potentially be considered for publication. However, 2 central points need further discussion:

1) Reconstruction

As pointed out also by the other reviewers, the procedure for reconstructing the PDO, which is presented as a key finding, was not sufficiently clear. Even now that it has been extended, it leaves important open question. In particular, more explanation is needed for the statement that the measurement of the same observables at two times allows for a "complete reconstruction of the reduced pseudo-state R23". This cannot be true in general, unless one already assumes that all the expectation values for unequal observables are zero. In general, travelling along an OTC might not correspond to an exact identity operation.

2) Linearity

It is claimed that the PDO approach does not introduced non-linearity because "any two PDOs [...] can be related to one another via a linear transformation". Since PDOs generalize standard density operators, the same must be true for the special case of density operators, yet D-CTC descriptions based on density operators do violate linearity. Hence, the reason why and how PDOs preserve linearity in the face of an OTC or CTC, as conjectured in the conclusion, must be a different one. Since this is a critical motivation for this study, this point has to be explained in more detail. This is also related to Reviewer 2's question regarding the relation to D-CTCs and P-CTCs, which warrants further discussion.

Reviewer #2 (Remarks to the Author):

I have read the revised manuscript 'Quantum Entanglement Near Open Timelike Curves: Theory and Experiment' by Marletto et al., Nature Communications manuscript number NCOMMS-18-16504A, the reports of the other referees, and the authors' reply. I believe that the authors have sufficiently responded to my comments on the paper, and I hope that my remarks may also be useful to them in future research on this topic. I therefore recommend this paper for publication in Nature Communications.

Reviewer #3 (Remarks to the Author):

The authors have addressed most of my previous remarks, and I recommend that the manuscript is published essentially as is.

I add two minor remarks that the authors could take into consideration.

1. In the abstract "experimental demonstration", should probably be replaced with "experiment".

2. An early reference (possibly the first) for the grandfather paradox is the short story Paradox by Charles Cloukey

Charles Cloukey, "Paradox", Amazing Stories Quarterly, Summer 1929, page 392

Reply to the reviewers' comments.

Reviewer's comments in boldface, our replies in normal font.

Reviewer #1 (Remarks to the Author):

The manuscript has improved from the previous version and could potentially be considered for publication. However, 2 central points need further discussion:

1) Reconstruction

As pointed out also by the other reviewers, the procedure for reconstructing the PDO, which is presented as a key finding, was not sufficiently clear. Even now that it has been extended, it leaves important open question. In particular, more explanation is needed for the statement that the measurement of the same observables at two times allows for a "complete reconstruction of the reduced pseudo-state R23". This cannot be true in general, unless one already assumes that all the expectation values for unequal observables are zero.

Yes – we thank the reviewer for having noted a point of possible misunderstanding in our explanation. What we meant was that the observables measured at two times are drawn from the same set $\{X, Y, Z\}$, not that we measure the same observable in succession. We have now changed the sentence in question to make it clearer (page 4, right column):

1) an ensemble where one measures on photon A the whole set of observables $\{X, Y, Z\}$ at time t_1 and the same set at time t_2 , including all possible cross-correlations between different observables. This provides the full reconstruction of the reduced pseudo-state $R_{23} = \frac{1}{4}(I + \Sigma_{23})$.

In general, travelling along an OTC might not correspond to an exact identity operation.

This is an interesting point, which we have now commented on in the text, noting that our approach can accommodate this case too (page 3, right column):

Note also the subtlety that the qubit entering an OTC could undergo some unitary transformation. This transformation would not change its being maximally entangled with the other qubit, so it could be incorporated in the description above by modifying the reduced state of Q1 and Q2 and of Q2 and Q3 to be different maximally entangled states. However, it still remains true that the qubit just before entering the OTC (Q2) and just after emerging from it (Q3) are two copies of the same qubit, which is why they can be described by the PDO R_{23} .

2) Linearity

It is claimed that the PDO approach does not introduce non-linearity because "any two PDOs [...] can be related to one another via a linear transformation".

Yes.

Since PDOs generalize standard density operators, the same must be true for the special case of density operators, yet D-CTC descriptions based on density operators do violate linearity.

Yes.

Hence, the reason why and how PDOs preserve linearity in the face of an OTC or CTC, as conjectured in the conclusion, must be a different one.

This is a tangential point to the manuscript, but it is nice to have the opportunity to comment on it, thereby explaining further the novelty of our proposal and adding to the motivation behind the experimental simulation, which we now discuss in the conclusions (see below). The formalism of quantum mechanics is linear in the sense that any map M that transforms ρ_1 to σ_1 and ρ_2 to σ_2 , and is compatible with quantum mechanics, gives $M(\alpha\rho_1 + \beta\rho_2) = \alpha M(\rho_1) + \beta M(\rho_2)$. If one insists that the states of qubits in chronology-violating regions are density operators, one has to require that the map describing the chronology-violating region is not linear, so the above property does not hold. This is the reason why CTCs and OTCs violate quantum mechanics.

By proposing to use a PDO to describe the three qubits in the chronology-violating region we depart from standard quantum mechanics, because we use a non-positive operator to describe the state of the qubits. Our

proposal hints at a new way of formulating quantum theory, where, to describe a physical system with a certain dynamics, one gives the PDO as a faithful description of that physical situation. We implicitly define a PDO as faithful if it correctly describes the correlations among observables in different qubits. Now, once one takes that step, is it still possible to preserve some notion of linearity even when describing situations where properties like entanglement monogamy are violated? We conjecture that the answer is yes – because any two PDOs describing such different physical situations (e.g. two OTCs with different initial states) can be related by a linear transformation. The physical meaning of this transformation is open for investigation, and this, together with the extension of this formalism to P-CTCs and D-CTCs, is an interesting possible development of our work. This notion of linearity is, however, different from the linearity of quantum mechanical evolution.

Since this is a critical motivation for this study, this point has to be explained in more detail. This is also related to Reviewer 2's question regarding the relation to D-CTCs and P-CTCs, which warrants further discussion.

True, we have added an explanation along the above lines (page 5, right column):

“By proposing to use a PDO to describe the three qubits in the chronology-violating region we depart from standard quantum mechanics, because we use a non-positive operator to describe the state of the qubits. Our proposal hints to a new way of formulating quantum theory, where, to describe a physical system with a certain dynamics, one gives the PDO as a faithful description of that physical situation. We implicitly define a PDO as faithful if it correctly describes the correlations between observables in different qubits. Now, once that step is taken, is it still possible to preserve some notion of linearity even when describing situations where properties like entanglement monogamy are violated? We conjecture that the answer is yes – because any two PDOs describing such different physical situations (e.g. two OTCs with different initial states) can be related by a linear transformation.”

Note also that the critical motivation of this study is to propose a different description of the OTCs where one can consistently maintain that monogamy of entanglement is violated. All of the above issues add to the relevance of our paper, because they highlight several interesting questions that our paper opens up.

Reviewer #2 (Remarks to the Author):

I have read the revised manuscript ‘‘Quantum Entanglement Near Open Timelike Curves: Theory and Experiment’ by Marletto et al., Nature Communications manuscript number NCOMMS-18-16504A, the reports of the other referees, and the authors’ reply. I believe that the authors have sufficiently responded to my comments on the paper, and I hope that my remarks may also be useful to them in future research on this topic. I therefore recommend this paper for publication in Nature Communications.

We thank the reviewer for the helpful comments.

Reviewer #3 (Remarks to the Author):

The authors have addressed most of my previous remarks, and I recommend that the manuscript is published essentially as is. I add two minor remarks that the authors could take into consideration.

We thank the reviewer for the comments and for the remarks, which we have addressed by adding a reference to the nice story he mentioned to us.

1. In the abstract "experimental demonstration", should probably be replaced with "experiment".

Yes, corrected.

2. An early reference (possibly the first) for the grandfather paradox is the short story Paradox by Charles Cloukey

Charles Cloukey, "Paradox", Amazing Stories Quarterly, Summer 1929, page 392

We have added a reference – and thanks for pointing this out to us, it is very interesting.

REVIEWERS' COMMENTS:

Reviewer #1 (Remarks to the Author):

I believe the authors have now sufficiently addressed the reviewers' comments and can recommend the manuscript for publication.

REVIEWERS' COMMENTS:

Reviewer #1 (Remarks to the Author):

I believe the authors have now sufficiently addressed the reviewers' comments and can recommend the manuscript for publication.

We thank the referee for the helpful comments and suggestions.